# Distribution and Risk Factors of Malaria in the Greater Accra Region in Ghana

**DOI:** 10.3390/ijerph191912006

**Published:** 2022-09-22

**Authors:** Koh Kawaguchi, Elorm Donkor, Aparna Lal, Matthew Kelly, Kinley Wangdi

**Affiliations:** 1Research School of Earth Sciences, Australian National University, Acton, Canberra, ACT 2601, Australia; 2Jockey Club School of Public Health and Primary Care, Chinese University of Hong Kong, Hong Kong SAR, China; 3National Centre for Epidemiology and Population Health, College of Health and Medicine, Australian National University, Acton, Canberra, ACT 2601, Australia

**Keywords:** Greater Accra region, Ghana, malaria, space, time, clustering, modelling

## Abstract

Malaria remains a serious public health challenge in Ghana including the Greater Accra Region. This study aimed to quantify the spatial, temporal and spatio-temporal patterns of malaria in the Greater Accra Region to inform targeted allocation of health resources. Malaria cases data from 2015 to 2019 were obtained from the Ghanaian District Health Information and Management System and aggregated at a district and monthly level. Spatial analysis was conducted using the Global Moran’s I, Getis-Ord Gi*, and local indicators of spatial autocorrelation. Kulldorff’s space–time scan statistics were used to investigate space–time clustering. A negative binomial regression was used to find correlations between climatic factors and sociodemographic characteristics and the incidence of malaria. A total of 1,105,370 malaria cases were reported between 2015 and 2019. Significant seasonal variation was observed, with June and July being the peak months of reported malaria cases. The hotspots districts were Kpone-Katamanso Municipal District, Ashaiman Municipal Districts, Tema Municipal District, and La-Nkwantanang-Madina Municipal District. While La-Nkwantanang-Madina Municipal District was high-high cluster. The Spatio-temporal clusters occurred between February 2015 and July 2017 in the districts of Ningo-Prampram, Shai-Osudoku, Ashaiman Municipal, and Kpone-Katamanso Municipal with a radius of 26.63 km and an relative risk of 4.66 (*p* < 0.001). Malaria cases were positively associated with monthly rainfall (adjusted odds ratio [AOR] = 1.01; 95% confidence interval [CI] = 1.005, 1.016) and the previous month’s cases (AOR = 1.064; 95% CI 1.062, 1.065) and negatively correlated with minimum temperature (AOR = 0.86, 95% CI = 0.823, 0.899) and population density (AOR = 0.996, 95% CI = 0.994, 0.998). Malaria control and prevention should be strengthened in hotspot districts in the appropriate months to improve program effectiveness.

## 1. Introduction

Malaria is a vector-borne tropical disease present in 85 countries worldwide, infecting approximately 241 million people, and causing 627,000 deaths in 2020 [1]. The age-standardized rate of disability-adjusted life years (DALYs) from malaria was 6498 per 100,000 population in 2017 [2]. In the same year, the World Health Organization (WHO) African Region reported 95% and 96% of global malaria cases and deaths, respectively [3]. Malaria is caused by *Plasmodium* parasites and is predominantly transmitted via the bite of the female *Anopheles* mosquito, but it can also be transmitted from mother to child or via blood transfusion [4]. While malaria is present in many regions, Africa and Asia carries significant burden and deaths [1,5,6]. Over 65% of the global death toll from malaria is in children under the age of five years, as repeated infections lead to near-complete immunity from severe disease and death, thereby protecting older people [7,8,9]. Malaria is endemic in Ghana, accounting for 40% of all outpatient visits to hospital [10].

Disease incidence is influenced by a wide range of factors, including the environmental suitability for the vectors, the population at risk, and the control measures taken. The primary malaria vectors in Ghana are *Anopheles gambiae* and *An. funestus*, with *An. gambiae* becoming more dominant as urbanization progresses, and *P. falciparum* is the primary *Plasmodium* parasite [11,12]. Factors that influence environmental suitability for mosquito vectors include climate, altitude, vegetation, and control measures [4]. *P. falciparum*, once transmitted to the female mosquito from an infected human, has a temperature-dependent extrinsic incubation period before becoming infectious to other individuals [13,14]. *An. gambiae* and *An. funestus* tend to be most prevalent in tropical, humid conditions with access to small pools of water [15,16]. Control measures such as sleeping under long-lasting insecticide nets (LLINs) can reduce malaria incidence significantly, with the WHO estimating that 1.7 billion cases of malaria were averted between 2000 and 2020 [1]. However, progress towards malaria elimination has stalled in recent years. Globally, malaria incidence declined by 27% between 2000 and 2015 but by only 1.2% between 2015 and 2019 [9]. In certain areas, the burden due to malaria is increasing due to resource limitations for health systems and drug and insecticide resistance as well as expanding climatic suitability [11].

Across Ghana, malaria incidence per 1000 people decreased by 7.8% each year between 2011 and 2018 but by only 2.6% per year between 2018 and 2020 [17]. To continue the success of control measures in Ghana, programs need to take into account the spatially heterogeneous nature of the malaria burden, as well as health care access [18,19,20]. In the Greater Accra Region, only 2% of children aged between 6–59 months tested positive for malaria by microscopy, compared with 27% in the western region of Ghana [21]. Therefore, targeted distribution of preventative measures is more likely to be effective as opposed to uniform black resource allocation [22]. Hotspot and cluster analyses are useful tools for assessing geographic variation in malaria risk and incidence [23,24,25,26]. Understanding the temporality of transmission can further support decisions regarding the timing of control measure deployment. Therefore, this study aimed to quantify the spatial, temporal and spatiotemporal patterns of malaria in the Greater Accra Region using retrospective surveillance data. The regional malaria program can use the findings from this study for targeted deployments of control measures including LLINs, indoor residual spraying (IRS) and intermittent preventive therapy in pregnancy (IPTp).

## 2. Materials and Methods

### 2.1. Study Site

The Greater Accra Region is the smallest of the 10 administrative regions in Ghana, spanning 3245 km^2^ in the southeast of the country (Figure 1) [27]. According to the 2021 census, it is the most populous region in Ghana with 5,455,692 people in the region [28]. Correspondingly, the population density in Greater Accra Region is much higher than in the rest of Ghana. Greater Accra Region has 29 metropolitan and municipal district assemblies (MMDAs), hence referred to as districts [29]. The MMDAs that are mentioned throughout this paper are: 1-Weija-Gbawe Municipal District, 2-Accra Metropolitan District, 3-Adenta Municipal District, 4-Ashaiman Municipal District, 6-Shai-Osudoku District, 7-Ga East Municipal District, 10-Tema Metropolitan District, 11-Ningo-Prampram District, 13-Ga North Municipal District, 17-Ga Central Municipal District, 18-La-Nkwantanang-Madina Municipal District, 19-Kpone-Katamanso Municipal District, 20-Ayawaso East Municipal District, 21-La-Dade-Kotopon Municipal District, 22-Korle-Klottey Municipal District, 23-Ayawaso West Municipal District, 24-Ayawaso North Municipal District, 25-Okaikwei North Municipal District, 26-Ayawaso Central Municipal District, 27-Ablekuma Central Municipal District, 28-Ablekuma North Municipal District and 29-Ablekuma West Municipal District (Figure 1). The Greater Accra Region has a tropical climate, with high humidity and average daily temperatures above 27 °C [30]. Variability in temperature is low, while the rainfall can vary significantly depending on the season [30].

### 2.2. Data Sources

This study used secondary aggregated clinical data for all age groups of confirmed malaria infections between 2015 and 2019, from hospitals and primary health facilities (hereafter referred to as health facilities) in the Greater Accra Region. These data were obtained from the Ghana Health Service and the Greater Accra Regional Health Directorate. Patients with fever or suspected malaria undergo blood tests with either a rapid diagnostic test (RDs) or microscopy. Malaria-positive patients are treated using the national guidelines, and their results are recorded by the treating health facilities. These data are aggregated temporally monthly and spatially at a district level and reported to the national surveillance system at the end of each month.

The district shapefiles were obtained from OpenStreetMap and imported into QGIS version 3.22 [31]. The yearly district population was obtained from the Health Information Department of the Greater Accra Regional Health Directorate, and the monthly population was linearly interpolated from yearly population data between 2015 and 2019. District-level monthly temperature and rainfall variables were determined based on data from WorldClim at a spatial resolution of 1 km [30].

### 2.3. Descriptive Statistics

Malaria data were aggregated across 2015–2019 and normalised for population by the Ghana Statistical Service (GSS) projected population in 2019. A choropleth map was created to represent the annual parasite index (API), which is the number of malaria cases each year per 1000 people [32]. Average monthly rainfall and maximum and minimum temperatures for each of the districts were calculated and are presented.

### 2.4. Temporal Analysis

The data were aggregated across districts to calculate monthly malaria cases in the Greater Accra Region. The seasonality of malaria was investigated using boxplots and line plots using monthly data. Means and 95% CIs for the data were considered for the line plots. Comparison between June and July mean reported cases to the rest of the year was conducted with Welch’s *t*-test.

### 2.5. Spatial Analysis

Spatial analysis was undertaken with Global Moran’s I, the Getis-Ord Gi* statistic, and local indicators of spatial autocorrelation (LISA) [33,34,35]. The Global Moran’s I statistic provides a measure of the spatial autocorrelation (tendency of neighbouring regions to have similar values), where a positive value indicates positive spatial autocorrelation, and a negative value indicates negative spatial autocorrelation [34]. Hotspot districts (Gi* statistic) are districts that have higher reported malaria cases than the mean for the Greater Accra Region [33]. LISA compares the similarity of risk between a district and its neighbours: a positive value indicates similar risk, while a negative value indicates dissimilar risk [35]. This can be used to obtain clusters (high risk surrounded by high risk), cold spots (low risk surrounded by low risk) and outliers (high risk surrounded by low risk and low risk surrounded by high risk) [35]. For example, a result of low-high represents a district with low risk surrounded by districts with high risk.

### 2.6. Space–Time Analysis

Kulldorff’s space–time statistic was used to identify likely clusters in space–time, that is, monthly by districts over the study period. Three sets of data were input into the SaTScan software: the monthly district malaria cases, the population in each of the 29 districts and the latitude and longitude of the centroids of each district. Kulldorff’s scan statistic utilises moving cylinders with the base of the cylinder in the spatial component and the height in the temporal component, with the maximum spatial cluster size set at 25% of the population at risk and the maximum temporal cluster size at 50% of the study period [36]. It considers the observed cases inside and outside the windows to estimate the likelihood that the observed cases inside the window are greater than what is expected (estimated as relative risk [RR]) by chance [36]. The window size with the highest log-likelihood ratio (LLR) was considered the most probable cluster, i.e., the cluster that is least likely to have occurred randomly. Statistical significance of clusters was obtained with 999 Monte Carlo replications at *p* <0.05.

### 2.7. Regression Analysis

Negative binomial regression was chosen over Poisson regression due to a lower Akaike information criterion (AIC) (Appendix A). A negative binomial generalized linear model was used to find the correlations between monthly malaria cases (Y) with the climatic covariates (rainfall, temperature), previous month’s malaria cases (log base 1.1 transformed) and population density (log base 1.1 transformed). A collinearity analysis using the variance inflation factor (VIF) showed that there was some collinearity between the maximum and minimum temperature (Appendix A). Therefore, two models with and without maximum temperature were fitted to select the best model using AIC. The model with minimum temperature, rainfall, malaria cases of the previous and population density was the model with the lowest AIC (Appendix A).

The model with the lowest AIC was a model that used the minimum temperature, rainfall, the previous month’s reported cases (log-transformed), and the population density (log-transformed).
(1)$$Y_{\mathrm{it}}\sim \mathrm{NegBin}\left(\mu_{\mathrm{it}},\;\theta \right)$$
(2)$$\log\left(\mu_{\mathrm{it}}\right)=\alpha +{\beta \mathrm{Tmin}}_{\mathrm{it}}+{\gamma \mathrm{Rainfall}}_{\mathrm{it}}+\delta \log_{1.1}\left(Y_{t-1}\right)+\epsilon \log_{1.1}\left({\mathrm{Popdens}}_{\mathrm{it}}\right)$$
where α, β, γ, δ, and ϵ are regression coefficients; Tmin_it_ and Rainfall_it_ are the minimum temperature and rainfall for district i at month t; Y_t−1_ is the previous month log-transformed and Popdens_it_ is the population density log-transformed.

Space–time analysis was performed with the SaTScan^TM^ software v10.0.2 (MA, USA). Temporal on Python using Spyder IDE (version 5.1.5). Regression analysis was conducted using Stata 16.1 (StataCorp LLC, TX, USA). Maps were created with ArcGIS v10.8.1 (ESRI, Redlands, CA, USA).

## 3. Results

### 3.1. Descriptive Statistics

Across the study period, a total of 1,105,370 malaria cases were recorded in the region. The eastern districts of the region had higher rates of malaria as shown in the choropleth map (Figure 1), while the districts near the Accra metropolitan district in the southwest tended to have low API (Figure 2). Average minimum temperatures ranged between districts from 23.3 °C to 24.4 °C (mean = 23.9 °C), while average maximum temperatures were between 31.3 °C and 32.4 °C (mean = 31.9 °C). Rainfall varied between districts from 75.8 mm to 96.2 mm per month, with the mean being 83.7 mm. A table detailing the total reported malaria cases, populations and the API and climate data can be found in Appendix A.

### 3.2. Temporal Analysis

Figure 2 shows that June and July represent clear peaks in the monthly case load in the Greater Accra Region, and cases tend to fluctuate in the other months, with an increase in one month tending to follow a decrease in the previous month. Larger interannual variation, indicated by a wider confidence interval (Figure 3), can be seen in the months with lower mean case numbers. In late 2017/early 2018 there was a clear period in which the incidence of malaria was substantially higher than the mean, and in the corresponding period in 2018/2019 the incidence was much lower than the mean. June and July both had mean reported case numbers higher than the rest of the year (*p* < 0.05), indicating that malaria prevalence is statistically significantly higher during these months.

### 3.3. Spatial Analysis

The Global Moran’s I was 0.111 (*p*-value < 0.001) indicating a clustering effect (Appendix A). From the Getis Ord Gi* analysis, the hotspot districts were Kpone-Katamanso Municipal District, Ashaiman Municipal District, Tema Municipal District, and La-Nkwantanang-Madina Municipal District (Figure 4A). Cluster analysis using LISA showed that only La-Nkwantanang-Madina Municipal District was a high–high hotspot. Tema Metropolitan and Accra Metropolis were low–high and high–low outliers, while Ayawaso West, La-Dade-Kotopon, Ayawaso North, Ayawaso East, Ayawaso Central, Korle-Klottey, Ablekuma West, Weija-Gbawe, Ablekuma North, Ablekuma Central, Okaikwei North and Ga Central districts are low–low cold spots (Figure 4B).

### 3.4. Space–Time Analysis

There were eight space–time clusters from SaTScan analysis. The most likely multi-district cluster occurred between February 2015 and July 2017 in Ningo-Prampram, Shai-Osudoku, Ashaiman Municipal and Kpone-Katamanso Municipal districts with a radius of 26.63 km and an RR of 4.66. A secondary cluster occurred in the districts of Ga East, Ga North and La-Nkwantanang-Madina between June 2017 and November 2019, with a radius of 7.94 km and a risk ratio of 1.96. Several of the clusters had a radius of 0 km, which indicates a single district (Figure 5 and Table 1).

### 3.5. Negative-Binomial Analysis

A 1 cm increase in monthly rainfall is associated with a 1.0% increase in malaria cases (relative risk [RR] = 1.01, 95% confidence interval [CI] 0.5%, 1.6%). A decrease in cases by 14.0% (RR = 0.87, 95% CI 10.1%, 17.7%) was associated with a 1 °C increase in minimum temperature. An increase in malaria cases by 10% in the previous month is correlated with an increase in malaria cases in the subsequent month by 6.4% (RR = 1.06, 95% CI 6.2%, 6.5%). A 10% increase in population density was correlated with a 0.4% lower incidence of reported malaria (RR = 0.996, 95% CI 0.2%, 0.4%) (Table 2).

## 4. Discussion

This analysis considered the variation in malaria burden between districts and over time across the Greater Accra Region, as well as identifying the climatic drivers of malaria in this region. There was significant inter-district variation of malaria incidence, with the API varying between 4 and 185. Temporal analysis showed a seasonal peak in June-July each year, with cases tending to increase and decrease in the other months. Kpone-Katamanso district reported higher than expected malaria during the study period. Rainfall and malaria in the previous month were positively associated with reported malaria, while temperature and population density were negatively associated with malaria.

Rainfall was positively correlated with reported malaria cases. This finding is similar to other studies which reported rainfall as an important driver of malaria transmission [37,38,39,40]. Rainfall provides breeding habitats for *An. gambiae* and *An. funestus*, as documented in the fringes of the North African deserts where *An. gambiae* and *An. funestus* populations increase rapidly at the onset of rain [6,41,42]. Therefore, people should be encouraged to implement control measures including sleeping under LLINs, particularly in months with high rainfall. Use of LLINs is particularly effective as most female mosquitos carrying *P. falciparum* only bite at night [43]. Further, febrile cases during the rainy season need investigation for prompt treatment with anti-malarial drugs and integrated vector control initiated as outlined in the Global Technical Strategy for Malaria 2016–2030 [44]. Vector control can be accomplished with LLINs and indoor residual spraying (IRS) and by reducing the number of breeding sites such as puddles caused by broken pipes or unfilled dugouts [12,44].

In this study, cases in the previous month were positively associated with the malaria case load of the current month. Malaria-infected individuals can pass on the infection to female *An. gambiae* and *An. funestus* through the blood meal, which in turn will infect healthy individuals in subsequent bites. Matured gametocytes (transmissible stage from humans to mosquitoes) are ingested by female mosquitoes and sporogonic cycle (in mosquitoes) of *P. falciparum* is around 11–16 days [45]. It takes around 10 days for *P. falciparum* to become gametocytes in humans [45]. Therefore, data from previous months can be used as a proxy for infection for the subsequent month. This information can be used for identifying control measures such as distributing LLINs and monitoring their use.

Minimum temperature was negatively correlated with malaria incidence. This result contrasts with the findings of Bi et al. (2003) in China and Mohammadkhani et al. (2016) in Iran, who found that minimum temperature was positively correlated with malaria incidence [40,46]. A study by Dabaro et al. (2021) in Ethiopia reported that minimum temperature was not a statistically significant covariate of malaria [47]. However, the research areas for these three studies were substantially colder than the Greater Accra Region, with the mean minimum temperature less than 20 °C in each case. By contrast, in Ghana, the minimum temperature was greater than 22.9 °C every month, and the maximum temperature was greater than 28.9 °C every month (Appendix A). This is significant as Wang et al. (2022) noted that the temperature-dependent reproduction number was maximised at around 25 °C [48]. It is hence likely that the mean temperature is greater than the optimal temperature for reproduction in the Greater Accra Region, which could be a reason for the negative correlation found. Another reason could be that the impact of temperature on malaria incidence due to *An. gambiae*, *An. funestus* and *P. falciparum* survival is complex and not fully captured in the model [49,50,51]. Patz and Olson (2006) noted that increases in temperature will lead to a decreased extrinsic incubation period of the parasite [52]. By contrast, Noden et al. (1995) found that particularly high temperatures reduce parasite densities and infection rates [50]. These findings can be reconciled by the model of Shapiro et al. (2017), which captured the observed biology of *P. falciparum* with *An. stephensi*, and found an optimum transmission temperature of 26 °C, with transmission possible in the temperature range between 17 °C and 35 °C [53]. These results may vary depending upon location, as Shapiro et al. (2017) used incubators for their experiments, and Blanford et al. (2013) found that using mean monthly temperature as an input into models will result in an overestimation of parasite development in warmer climates such as Ghana due to effects of diurnal variation [14,53]. Variation in temperature also affects the biting rate and gonotrophic processes [54,55].

Population density was weakly negatively correlated with malaria cases. This agrees with a large amount of the literature regarding malaria incidence in urbanised areas. Several studies have indicated that urban areas have lower levels of malaria incidence [56,57]. Kabaria et al. (2017) found that malaria risk and population density follow a nonlinear relationship, with malaria risk positively correlated with population densities smaller than 1000 people/km^2^ and negatively correlated with population densities greater than 1000 people/km^2^ [58]. In the Greater Accra Region, most of the districts have a population density greater than this value, so this agrees with the findings. There are several reasons why this is the case. Residents in urban areas in the Greater Accra Region have increased access to houses which are made of materials less susceptible to mosquitos entering the home [59]. Wealthier urban areas are associated with lower incidence of malaria due to increased access to health services, better garbage collection, higher sewer connection rate, and less open water sites susceptible to mosquito breeding [16,60,61]. However, *An. gambiae* adapting to breed in polluted regions and urban areas having lower uptake of malaria prevention measures in Ghana may provide a reason why the correlation was weak [12,21,62].

Malaria in the Greater Accra Region is highly seasonal, with cases peaking around June and July every year. Mattah et al. (2017) found that around a third of the sampled *An. gambiae* habitats in Ghana dried up at least once in an 11-month period [12]. Correspondingly, *An. gambiae* and *An. funestus* habitats are more numerous in the wet season, which may be a driver of high malaria case numbers (Appendix A). Another driver may be that the weeding of major crops such as corn occurs during this season, providing more vector–human interactions, which drive malaria transmission [63]. Large interannual variation was observed in the months with fewer mean reported cases. This interannual variation may be climate driven, with large systems such as the Atlantic Multidecadal Oscillation and the Inter-decadal Pacific Oscillations being drivers of rainfall in West Africa [64]. Another potential reason for this result is malaria control measures such as LLIN and IRS that are in place in Ghana [21]. Grassly and Fraser used a theoretical model of a general seasonal disease to show that control measures can lead to more variability in cases [65].

The space–time model found eight clusters in the observation period, and the observed number of cases in these districts was statistically significantly higher than the expected number of cases. The primary cluster identified from Kulldorff’s space–time scan statistic covered the Ningo-Prampram, Shai-Osudoku, Ashaiman Municipal and Kpone-Katamanso Municipal districts in the centre of the Greater Accra Region between February 2015 and July 2017. These districts were not significant in Getis-Ord Gi* and LISA. This may be because these larger districts do not have centroids near other districts, leading to wider confidence intervals [33]. La-Nkwantanang-Madina, which was the hotspot identified, was part of the second most likely cluster, and Tema Metropolitan District, which borders the most likely cluster and the single-district clusters was classed to be a low-high outlier. According to the 2021 Ghana census, districts in the primary space–time cluster tend to be rural, with Shai-Osudoku and Ningo-Prampram with low population density compared with the average across the Greater Accra Region [28] (Appendix A). This agrees with the existing literature regarding increased transmission intensity in rural areas due to the *An. gambiae* vector’s lack of fitness for urban settings/fitness [16]. The consistency of these findings using different analytical methods provides confidence in the existence of hotspots in particular locations. Therefore, spatially targeted resource allocation is likely to improve the cost effectiveness of control programs.

There are several limitations to this study. First, the most important limitation of the current study is the potential for the inconsistent reporting of malaria across health centres and the likelihood that not all cases were captured by the passive surveillance system. Second, populations of districts were projected and may have led to under- or overestimation. Third, unmeasured risk modifiers such as living standards and socioeconomic status of the study population, localised behavioural and treatment seeking patterns, population mobility, LLIN use and residual indoor insecticide coverage were unaccounted for in the model. Future studies should consider a wider range of potential covariates for a better understanding of the impacts of relationships between environmental conditions and malaria incidence.

Despite these limitations, this study provides key insights into the districts within the Greater Accra Region which have higher cases of reported malaria and can be used to prioritize resources into districts with more cases. Interventions can be targeted around June and July when the reported cases peak. In addition, the regression analysis, while missing some risk modifiers, captures key climatic risk factors such as minimum temperature and rainfall that were associated with malaria incidence.

## 5. Conclusions

Malaria cases were clustered in the central district in the Greater Accra Region in Ghana and incidences were higher in June and July months. Climatic and sociodemographic were also associated with the transmission of malaria in this study. The spatial heterogeneity of malaria incidence even within a small region such as Greater Accra highlights the need to identify high-risk areas (districts) for targeted interventions. Therefore, national malaria program should prioritize these hotspot districts for malaria control activities including LLINs, IRS and IPTp.

## Figures and Tables

**Figure 1 ijerph-19-12006-f001:**
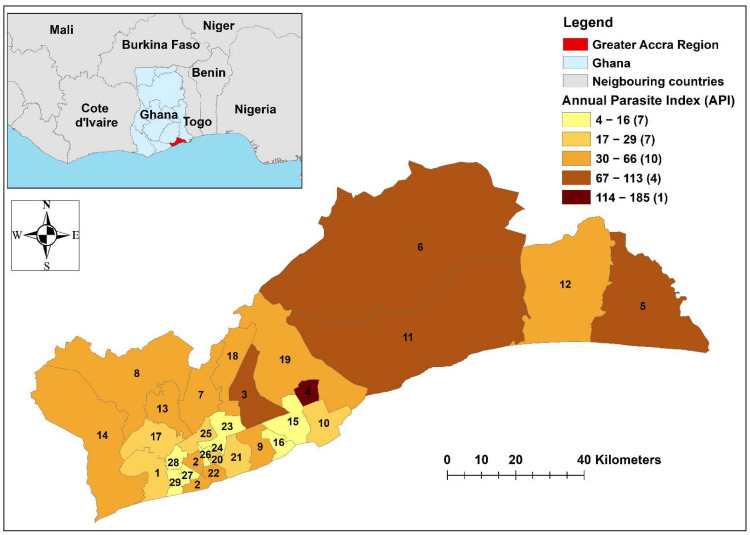
Map of study area (inset) and a choropleth map of the Greater Accra Region indicating the mean number of malaria infections per year (per 1000 people), 2015–2019.

**Figure 2 ijerph-19-12006-f002:**
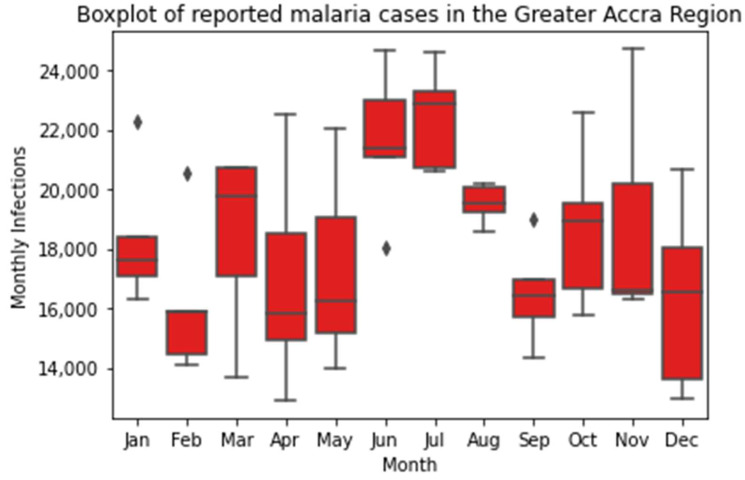
Boxplot of monthly reported malaria cases in the Greater Accra Region, 2015–2019. Diamonds indicate outliers.

**Figure 3 ijerph-19-12006-f003:**
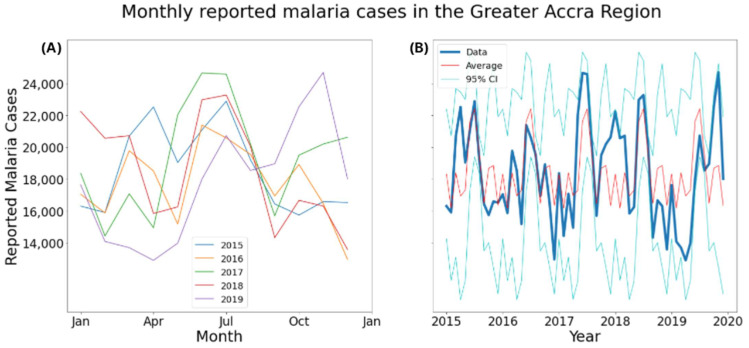
Line plot of monthly reported malaria cases with mean and 95% confidence interval in the Greater Accra Region, 2015–2019: (**A**) Each year in different colours; (**B**) Whole study period as one line.

**Figure 4 ijerph-19-12006-f004:**
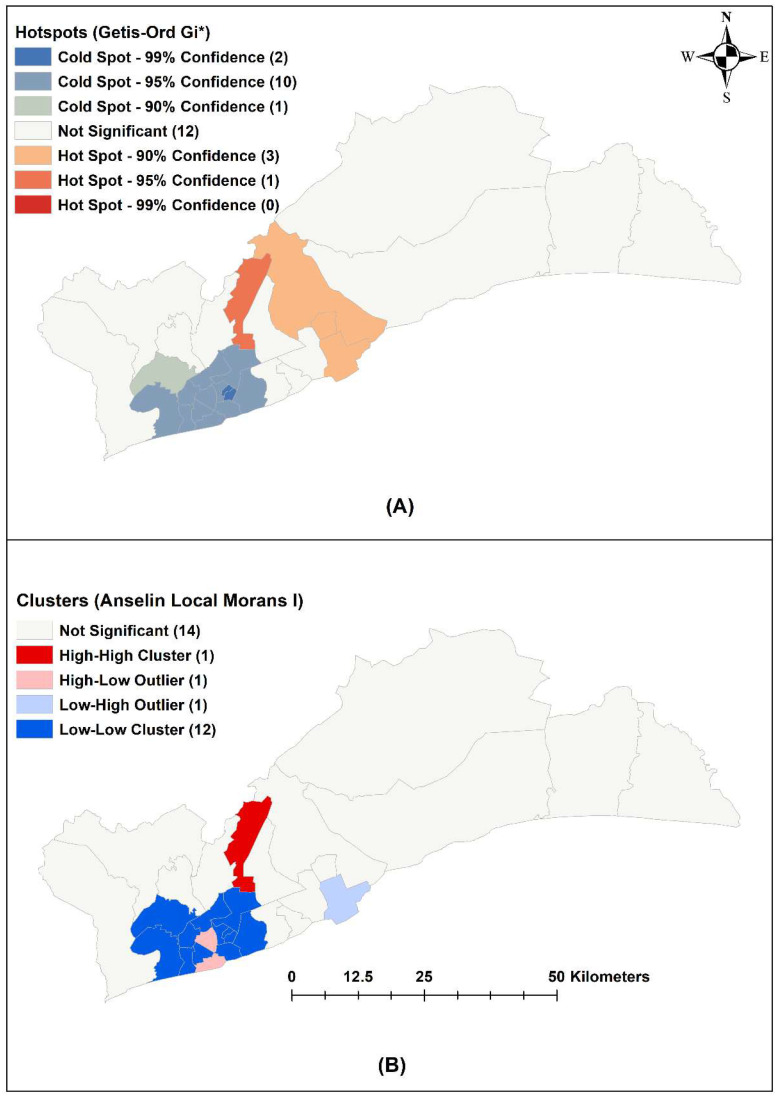
Hotspot analysis of reported malaria cases in the Greater Accra Region, 2015–2019: (**A**) Getis-Ord Gi* statistic; (**B**) LISA—HH is high risk surrounded by high risk, LH is low risk surrounded by high risk.

**Figure 5 ijerph-19-12006-f005:**
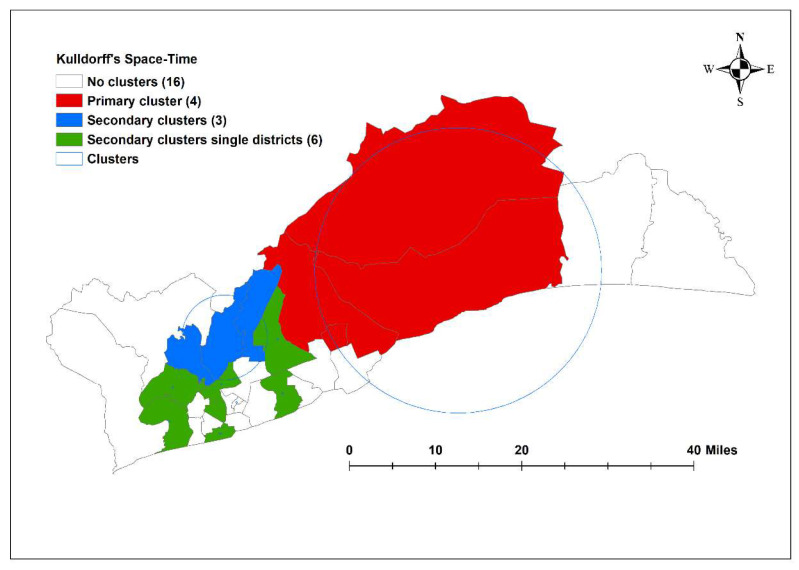
A significantly high rate of spatio-temporal malaria clusters with Kulldorff’s space–time scan statistic in the Greater Accra Region, 2015–2019.

**Table 1 ijerph-19-12006-t001:** The date range and location of space–time clusters of malaria incidence in the Greater Accra Region, 2015–2019.

Time Period (Month Year)	Latitude	Longitude	Radius (km)	Population	No. of Cases	Expected Cases	No. of Districts	RR	LLR	*p*-Value *
15 February–17 July	5.8126	0.1730	26.6	483,466	216,521	54,874.3	4	4.66	148,688.1	<0.001
15 February–17 July	5.6967	0.1308	0	88,105	41,972	10,000.12	1	4.32	28,704.2	<0.001
15 May–16 November	5.6060	0.1218	0	156,336	15,921	11,208.48	1	1.43	885.4	<0.001
15 July	5.5386	0.2269	0	509,868	2243	1918.16	1	1.17	26.1	<0.001
17 June–19 November	5.7000	0.2196	7.9	411,736	94,709	50,348.11	3	1.96	16,426.8	<0.001
17 June–19 November	5.6169	0.3065	0	133,495	22,665	16,324.08	1	1.40	1115.8	<0.001
18 April	5.5591	0.3128	0	180,058	849	714.82	1	1.19	11.9	<0.05
19 August–19 November	5.6212	0.2249	0	236,067	8296	3990.12	1	2.09	1774.8	<0.001

RR—relative risk; LLR—log-likelihood ratio; * significant at *p* value of 0.05.

**Table 2 ijerph-19-12006-t002:** Associations between environmental variables and malaria incidence risk in the Greater Accra Region, 2015–2019.

Covariates	Coefficient	95% CI	Risk Ratio	95% CI
Intercept	5.610	5.001, 6.218	−	−
Rainfall (cm)	$1.013\times 10^{-2}$	$7.29\times 10^{-3},\;1.297\times 10^{-2}$	1.01	1.005, 1.016
Min temp (°C)	−0.137	−0.161,−0.113	0.86	0.823, 0.899
Malaria cases *	$6.17\times 10^{-2}$	$5.99\times 10^{-2},\;6.34\times 10^{-2}$	1.064	1.062, 1.065
Population density **	$-3.69\times 10^{-3}$	$-5.55\times 10^{-3},-1.83\times 10^{-3}$	0.996	0.994, 0.998

CI—confidence interval; * Malaria cases of previous month log transformed; ** Population density log transformed.

## Data Availability

The datasets generated and/or analyzed during the current study are available from the corresponding author on reasonable request.

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
