# Peer review of "Distribution and Risk Factors of Malaria in the Greater Accra Region in Ghana"

_ijerph, 2022, doi:10.3390/ijerph191912006_

Round 1
Reviewer 1 Report
Minor comments are attached.

Author Response
Reviewer 1
The suggestions are the following:
- Please provide either a detailed flowchart or a map to explain the timeline and secondary data collected for each year.
Response: Thanks for the suggestion to add the timeline of data collection. However, we are unable to provide this information since we got the data from the Ghana Health Service and Greater Accra Regional Health Directorate (line numbers 112-113).
- Combine Figures 1 and 2 into a panel, it will improve readability.
Response: Thanks for the suggestion. We have combined the two maps.
- Figure 3 Caption indicates “Boxplot of region-wide malaria incidence”, but in the graph, there is no mention of the regions. Either improve the caption or change the graph, otherwise, Figure 3 and 4(a) confuses each other.
Response: We have changed “region-wide” to “the Greater Accra Region” (page number 5, line number 206).
- Also, in Figure 3 or its legends, please mention if the infection in June and July is significantly (p<0.05) higher than in other months. It will be a nice take-home message for readers.
Response: The methods and results have been updated to show the methods of significance.
Page number 4, line number 137-138:
Comparison of June and July mean reported cases to the rest of the year was conducted by Welch's t-test.
Page 5, line numbers 206-208:
June and July both had mean reported case numbers greater than the rest of the year (p<0.05), indicating malaria prevalence is statistically significantly higher during these months.
- The quality of Figure 4 (both a and b) are very poor, providing better quality. Also, I suggest taking out the average and 95% CI lines, it increases complexity and difficult to read.
Response: Figure 4 has been updated (note also have used monthly reported cases rather than incidence to more accurately reflect the data) – as some of the analysis provided in the results does rely on the mean and CI, has only been removed from figure 4A, rather than all of figure 4.

Reviewer 2 Report
Distribution and drivers of malaria in the Greater Accra Region 2 in Ghana
This is a very interesting manuscript, which deserves to be published at the IJERPH. I have only some comments to improve its presentation.
General comments:
♠ Even when written clearly, I think this Ms could be helped by a review by an English language expert.
♠ Check that all scientific names are correctly italicized.
♠ Instead of “mosquito” or “Anophdeles”, use the species (not just the genus) of each mosquito throughout the text (not just at the end of the Discussion), because the vectorial capacity of each species is different.
Specific comments:
♠ “Abstract: “This study aimed t to quantify…”. Please remove the “t”.
♠ Line 30. ….” and negatively correlated with minimum temperature (AOR= 0.86, 95% CI= 0.823, 0.899) and population density…”. Be careful with this relationship, because even when the statistical test supports your claim, the explanation provided in the Discussion is weak.
♠ Introduction. Lines 36-37. “Malaria is a vector-borne tropical disease present in 91 countries worldwide, infecting approximately 241 million people and causing 627,000 deaths in 2020 [1, 2].”. Please use the most recent source of information available, preferably the WHO.
https://www.who.int/news-room/fact-sheets/detail/malaria
♠ Line 42. Please delete reference No. 4, because it is too old (eleven years) and out of date.
♠ Lines 84-84. Great Accra has 29 metropolitan and municipal district assemblies (MMDAs), , hence referrred to as districts [26]. These MMSAs are…
I think it' is unnecessary to include all those MMSAs. Include only the most relevant to your study.
♠ Lines 102-109. 2.2. Data Sources. It is really necessary to include some supporting references to that various information.
♠ Discussion. “This is significant as Wang et al. noted that…” Please add the year of the reference in every case.
♠ Lines 302-310. This is the weak part of the manuscript, since the negative correlation between malaria cases and population density is indirect. That association is only supported by one study (Reference 13), whose data are open to interpretation. I suggest looking for more related references to enrich this part of the Discussion.
Author Response
Reviewer 2
General comments:
- Even when written clearly, I think this Ms could be helped by a review by an English language expert.
Response: A native English speaker academic has edited the manuscript.
- Check that all scientific names are correctly italicized.
Response: The scientific names have been italicised in the revised manuscript.
- Instead of “mosquito” or “Anopheles”, use the species (not just the genus) of each mosquito throughout the text (not just at the end of the Discussion), because the vectorial capacity of each species is different.
Response: The suggested changes have done in the revised manuscript. In some locations mosquito has been kept where it is unambiguous and makes the text flow better.
Specific comments:
- “Abstract: “This study aimed t to quantify…”. Please remove the “t”.
Response: We have made the correction as suggested.
- Line 30. ….” and negatively correlated with minimum temperature (AOR= 0.86, 95% CI= 0.823, 0.899) and population density…”. Be careful with this relationship, because even when the statistical test supports your claim, the explanation provided in the Discussion is weak.
Response: The discussion has been strengthened as follows.
Page number 10, Line numbers 320-337:
Population density was weakly negatively correlated with malaria cases. This agrees with a large amount of the literature regarding malaria incidence in urbanised areas. Several studies have indicated that urban areas have lower levels of malaria incidence [56, 57]. Kabaria et al. (2017) found that malaria risk and population density follow a non-linear relationship, with malaria risk positively correlated for population densities smaller than 1000 people/km2, and negatively correlated for population densities greater than 1000 people/km2 [58]. In the Greater Accra Region, most of the districts have a population density greater than this value so this agrees with the findings. There are several reasons why this is the case. Residents in urban areas in the Greater Accra Region have increased access to houses which are made of materials less susceptible to mosquitos entering the home [59]. Wealthier urban areas are associated with lower incidence of malaria, due to increased access to health services, better garbage collection, higher sewer connection rate, and less open water sites susceptible to mosquito breeding [16, 60, 61]. However, An. gambiae adapting to breed in polluted regions and urban areas having lower uptake of malaria prevention measures in Ghana may provide a reason why the correlation was weak [12, 21, 62].
- Lines 36-37. “Malaria is a vector-borne tropical disease present in 91 countries worldwide, infecting approximately 241 million people and causing 627,000 deaths in 2020 [1, 2].”. Please use the most recent source of information available, preferably the WHO. https://www.who.int/news-room/fact-sheets/detail/malaria
Response: The suggested updated have been made in the revised manuscript.
Page number 1, line numbers 36-40:
Malaria is a vector-borne tropical disease present in 85 countries worldwide, infecting approximately 241 million people and causing 627,000 deaths in 2020 [1]. The age-standardized rates of disability-adjusted life years (DALYs) rate from malaria was 649·8 per 100,000 populations [2]. World Health Organization (WHO) African Region reported 95% and 96% of malaria cases and deaths, respectively [3].
- Line 42. Please delete reference No. 4, because it is too old (eleven years) and out of date.
Response: Deleted, however this reference has been used to note that primary vectors of disease are An.gambiae and An.funestus (lines 48-49)
- Lines 84-84. Great Accra has 29 metropolitan and municipal district assemblies (MMDAs), , hence referrred to as districts [26]. These MMSAs are…
I think it' is unnecessary to include all those MMSAs. Include only the most relevant to your study.
Response: Have removed all of the names which are not mentioned further in the study, but this leaves a significant number of MMDAs left.
- Lines 102-109. 2.2. Data Sources. It is really necessary to include some supporting references to that various information.
Response: Addition of sentence in line 105-106.
- “This is significant as Wang et al. noted that…” Please add the year of the reference in every case.
Response: The year after the authors have been added in the revised manuscript.
- Lines 302-310. This is the weak part of the manuscript, since the negative correlation between malaria cases and population density is indirect. That association is only supported by one study (Reference 13), whose data are open to interpretation. I suggest looking for more related references to enrich this part of the Discussion.
Response: We have added additional arguments in the discussion to support the findings of negative binomial regression.
Page number 10, line numbers 320-337:
Population density was weakly negatively correlated with malaria cases. This agrees with a large amount of the literature regarding malaria incidence in urbanised areas. Several studies have indicated that urban areas have lower levels of malaria incidence [55, 56]. Kabaria et al. (2017) found that malaria risk and population density follow a non-linear relationship, with malaria risk positively correlated for population densities smaller than 1000 people/km2, and negatively correlated for population densities greater than 1000 people/km2 [57]. In the Greater Accra Region, most of the districts have a population density greater than this value so this agrees with the findings. There are several reasons why this is the case. Residents in urban areas in the Greater Accra Region have increased access to houses which are made of materials less susceptible to mosquitos entering the home [58]. Wealthier urban areas are associated with lower incidence of malaria, due to increased access to health services, better garbage collection, higher sewer connection rate, and less open water sites susceptible to mosquito breeding [15, 59, 60]. However, An. gambiae adapting to breed in polluted regions and urban areas having lower uptake of malaria prevention measures in Ghana may provide a reason why the correlation was weak [11, 20, 61].

Reviewer 3 Report
This is a very important topic, and readers of the fiend and general people as well will be interested. The manuscript is well structured and well written. However, describe your results in a greater detail. Sub-heading of the results are not suitable for this section. Please revise them to make interesting.
In introduction, please incorporate recent DALY.
In title replace the term 'divers' with 'risk factors'
Author Response
Reviewer 3
- In introduction, please incorporate recent DALY.
Response: We have incorporated DALYs due to malaria in the revised manuscript.
Page number 1, line numbers 37-38:
The age-standardized rates of disability-adjusted life years (DALYs) rate from malaria was 649·8 per 100,000 populations in 2017 [2].
- In title replace the term 'divers' with 'risk factors'.
Response: The title has been revised by changing “drivers” with “risk factors”.
